# Assessing Vaccination Delivery Strategies for Zero-Dose and Under-Immunized Children in the Fragile Context of Somalia

**DOI:** 10.3390/vaccines12020154

**Published:** 2024-02-01

**Authors:** Ahmed Said Bile, Mohamed A. Ali-Salad, Amina J. Mahmoud, Neha S. Singh, Nada Abdelmagid, Majdi M. Sabahelzain, Francesco Checchi, Sandra Mounier-Jack, Barni Nor

**Affiliations:** 1Somali Institute for Development Research and Analysis (SIDRA), Garowe, Puntland State, Somalia; mohamed.abdullahi@sidrainstitute.org (M.A.A.-S.); amina.jama@uu.se (A.J.M.); 2Department of Women’s and Children’s Health, Uppsala University, 753 10 Uppsala, Sweden; barni.nor@uu.se; 3Department of Global Health and Development, Faculty of Public Health and Policy, London School of Hygiene & Tropical Medicine, London WC1E 7HT, UK; neha.singh@lshtm.ac.uk (N.S.S.); nada.abdelmagid@lshtm.ac.uk (N.A.); francesco.checchi@lshtm.ac.uk (F.C.); 4Health in Humanitarian Crises Centre, London School of Hygiene & Tropical Medicine, London WC1E 7HT, UK; 5School of Health Sciences, Ahfad University for Women (AUW), Omdurman P.O. Box 167, Sudan; majdi.dafallah@sydney.edu.au; 6School of Public Health, Faculty of Medicine and Health, University of Sydney, Sydney, NSW 2050, Australia; 7Department of Infectious Disease Epidemiology and International Health, Faculty of Epidemiology and Population Health, London School of Hygiene & Tropical Medicine, London WC1E 7HT, UK; sandra.mounier-jack@lshtm.ac.uk

**Keywords:** vaccination, zero-dose children, populations at risk, crisis, conflict, fragility

## Abstract

Somalia is one of 20 countries in the world with the highest numbers of zero-dose children. This study aims to identify who and where zero-dose and under-vaccinated children are and what the existing vaccine delivery strategies to reach zero-dose children in Somalia are. This qualitative study was conducted in three geographically diverse regions of Somalia (rural/remote, nomadic/pastoralists, IDPs, and urban poor population), with government officials and NGO staff (*n* = 17), and with vaccinators and community members (*n* = 52). The data were analyzed using the GAVI Vaccine Alliance IRMMA framework. Nomadic populations, internally displaced persons, and populations living in remote and Al-shabaab-controlled areas are three vulnerable and neglected populations with a high proportion of zero-dose children. Despite the contextual heterogeneity of these population groups, the lack of targeted, population-specific strategies and meaningful engagement of local communities in the planning and implementation of immunization services is problematic in effectively reaching zero-dose children. This is, to our knowledge, the first study that examines vaccination strategies for zero-dose and under-vaccinated populations in the fragile context of Somalia. Evidence on populations at risk of vaccine-preventable diseases and barriers to vital vaccination services remain critical and urgent, especially in a country like Somalia with complex health system challenges.

## 1. Introduction

Childhood vaccination is one of the most significant and cost-effective global health inventions which saves millions of lives every year. Yet, more than 18 million children, most of whom live in low-income, fragile, and crisis-affected settings, were reported in 2021 to have not received any dose of the three basic childhood vaccinations, diphtheria, tetanus, and pertussis (DTP) vaccines [1]. In the 2030 Immunization Agenda, the United Nations set a target goal of achieving 90 percent coverage for essential vaccines for children and adolescents and halving the number of children completely missing out on vaccines, so-called zero-dose children [2]. If this target is met, an estimated 51.5 million vaccine-preventable deaths could be averted against 14 selected pathogens [3]. However, a recent analysis by the Global Burden of Diseases (GBD) shows that only 11 countries have achieved this target for all assessed vaccines [4].

Despite renewed global efforts to boost vaccination coverage, vaccine inequity persists, and many low-income countries continue to be left behind in accessing vital vaccines. Somalia is one of the 20 countries with the highest rates of zero-dose children, child morbidity, and mortality [1,5,6]. The country’s fragile and fragmented health system, unclear governance structure, and limited health financing hampers the routine access and delivery of essential vaccines [7]. The recurrent humanitarian health emergencies afflicting the country every four to five years further weaken the system by adding to the caseload of vector- and water-borne diseases and severe malnutrition of already vulnerable populations [8,9,10].

The reported low vaccination coverage in Somalia is derived from a combination of population estimates and monthly administrative immunization data from the health facilities. This is due to the weak health information system in the country and the lack of systematic and accurate data on catchment populations, given that a significant proportion of the Somali population is nomadic and periodically on the move [11,12]. The 2020 Somalia Health and Demographic Survey (SHDS) was the first survey in 30 years to collect demographic and health data. The data show that over 60 percent of Somali children did not receive any single dose of vaccines and are thus zero-dose children, while less than 11 percent of children between 12 and 23 months of age were fully vaccinated [13]. The survey was not conducted in large areas in South-central Somalia, which is largely controlled by Al-Shabaab. This may potentially indicate that the prevalence of unvaccinated in those hard-to-reach areas could be even higher than that of the rest of the country. This disparity and geographical variation in vaccination coverage in different regions of Somalia is symptomatic of the fragile and fragmented health system of the country. 

According to the recently revised national immunization policy and the Expanded Programme on Immunization (EPI), all children in Somalia under two years of age should receive vaccination against tuberculosis, diphtheria, tetanus, pertussis, polio, meningitis diseases caused by Haemophilus influenzae type B (Hib), measles, and hepatitis B [14]. However, recent data from the World Health Organization and the United Nations Children’s Fund show that only 37 percent of the target children in Somalia were vaccinated against BCG, while 42 percent received their third dose of DTP-containing vaccine [15]. While these statistics report childhood vaccination for the whole country, there are significant variances in the vaccination coverage among the different regions and communities.

The current knowledge gap on zero-dose children in Somalia makes national efforts to increase vaccine coverage difficult. This study therefore aims to contribute to new knowledge on who and where zero-dose children and under-vaccinated populations live in Somalia and what vaccine delivery strategies and mechanisms are in place. The study is part of a larger research project on zero-dose children in crisis-affected populations and uses the GAVI Vaccine Alliance IRMMA Framework to better understand zero-dose and missed communities in the fragile health system of Somalia [16,17]. 

## 2. Methods and Materials

### 2.1. Study Setting

The study was conducted in three geographically different regions (Puntland, Galmudug, and Jubbaland) in Somalia, which may be illustrative of the diverse context and population (rural/remote, nomadic/pastoralists, IDPs, and urban poor population) across Somalia. The three regions all have varying degrees of vulnerabilities and recurrent humanitarian emergencies. 

Puntland, situated in the northeast of Somalia, is the largest and relatively most stable region in Somalia. There are no accurate data on the number of people living in the region, but the last population survey estimated approximately 2 million people [11]. The region contains densely populated cities and “hard-to-reach” remote coastal areas as well as sparsely populated districts. A large proportion of the population living in the State are nomads and internally displaced people (IDPs). Other communities in the State include fishing households living in the coastal towns along the Gulf of Aden and the Indian Ocean. 

Galmudug, located in central Somalia, is the smallest State in Somalia. The State was formally established as a Federal Member State in 2015 and shares borders with Puntland State as well as the Somali region of Ethiopia. Its population is estimated to be 1.3 million and contains a large urban and nomadic population (44% and 31%, respectively) [18]. Similar to Jubbaland, the State suffered from prolonged conflict with Al-shabaab militants and poor infrastructure, which makes some of the coastal areas along the Indian Ocean “hard-to-reach” areas. 

Jubbaland is a densely populated State situated in Southern Somalia and was formally established as a Federal Member State in 2013. The State contains the coastal city of Kismayo, several towns along the border with Kenya and Ethiopia, and large fertile agricultural areas along the Jubba River. The population, estimated to be 1.36 million, consists of large sedentary communities (urban, farming, and fishing communities), pastoralists, internally displaced people, and refugee returnees [19]. Insecurity due to the presence of Al-shabaab militants and poor infrastructure make large parts of the State “hard-to-reach” areas. 

### 2.2. Study Design

The study used qualitative methods to understand who and where zero-dose and under-immunized children were and explore reasons for why these children may be left behind. The chosen qualitative design is assessed to be appropriate in understanding people’s experiences of the access and utilization of vaccine services and the potential unmet needs. The study design provides additional information and thus complements the quantitative secondary data analysis of the 2020 SHDS data performed by the study team and submitted for publication separately. 

### 2.3. Sampling

Senior officials from the MoHs, EPI policymakers, and humanitarian workers who were responsible for vaccination service programming and delivery in the study sites were purposively selected for the key informant interviews. Six health facilities were identified as suitable recruitment sites for focus group discussions with vaccinators. The selection of these facilities was facilitated by the district health authorities using the criteria of location (close proximity to IDP population; deprived areas), level (health centre), and services (EPHS; vaccination outreach programmes). Vaccinators who were involved in administrating vaccines in the 6 health facilities were approached to take part in the study. Community members (caregivers) were also approached to participate in the focus group discussions. Oral and written consent were taken from the participants prior to the data collection. 

### 2.4. Data Collection

We developed topic guides for key informant interviews (KII) and focus group discussions (FGD) using the Gavi IRMMA framework. Between July 2022 and March 2023, we conducted 17 key informant interviews with MoH officials and NGO staff to understand who and where zero-dose children and missed communities were, explore the main barriers that were hindering the zero-dose children getting vaccinated, and identify the existing strategies and interventions to reach these communities in Somalia. Two researchers conducted twelve KII in-person and the other five KII remotely via Zoom. The qualitative interviews sought to assess the background and demographic information about zero-dose children and missed communities by exploring and reflecting the knowledge, understanding and experience of the participants [20]. 

Vaccinators from six health facilities participated in three FGDs (one FGD in each study site) between December 2022 and February 2023. Focus group discussions for the community members (caregivers) were conducted in two sites (Galmudug and Jubbaland) in December 2023. The FGD was used to encourage participants to express ideas and experiences that could reveal dimensions of understanding and meanings about the key health and social characteristics of zero-dose children and missed communities, which could not have been fully explored in the key informant interviews [21]. 

All KII and FGD were conducted in the Somali language by four bilingual Somali researchers who were involved in the study protocol and topic guide development. Although the KII and FGD were audio-recorded, field notes were also taken to document observations during the discussions in KII and FGD by one of the researchers. The audio-recordings of the KII and FGD were transcribed in Somali language by two researchers and were peer-checked by another two researchers. Four researchers undertook the English language translation and review of the transcripts and the field notes. 

### 2.5. Data Analysis

We analyzed the data thematically using the Gavi IRMMA framework [17,22]. The framework comprises five steps: Identify, Reach, Monitor, Measure, and Advocate (Table 1).

The KII and FGD data were coded and analyzed using Taguette software (2021, The Open Journal, www.taguette.org/about.html, (accessed on 20 December 2023)) [23]. The emerging themes and subthemes were mapped to the IRMMA framework and synthesized to provide a detailed description of zero-dose children and missed communities in Somalia [24]. 

The data were analyzed in English language. We acknowledge the possible effect of translation in cross-language data analysis and reporting, but rigorous quality checking was carried out to ensure that the originally intended meaning and emphasis were not altered.

## 3. Results

The results are presented according to themes in the GAVI Vaccine Alliance IRMMA framework (Figure 1). All of the 17 KII respondents were male, and most of them (65%, *n* = 11) worked for the government, while the remaining participants worked either for the United Nations or non-governmental organizations (NGOs) (Table 2). In comparison to the KII, the majority of the participants (83%, *n* = 43) in the FGD were female (Table 3).

### 3.1. Theme 1: Identification of Zero-Dose Children and Missed Communities

The lack of systematic and accurate population health data coupled with the limited access, availability and utilization of health services in Somalia makes it difficult, if not impossible, to identify, plan and effectively reach zero-dose children and under-immunized communities. An MoH official described the challenges in identifying and locating unvaccinated children and their communities.

*“There is insufficient data on the number of children under the age of one who reside even in the areas where health facilities are located because the routine immunization coverage reporting cannot be used to estimate the number of unvaccinated children as these children may have never been in contact with the health system. It is difficult to plan for population whose size and numbers are unknown. Large communities live in remote and inaccessible areas or places that cannot be visited due to insecurity”*.(MoH official, KII)

The adequate access and utilization of health services is not only a concern for rural and remote population groups but also one which affects densely populated communities in large cities. 

*“Zero dose children live everywhere in the country. The basis of my argument is related to the limited access to health services particularly immunization services in many places. If the mother does not have access to a health facility or immunization services (delivered as an outreach or mobile), it is highly likely that the (her) child will be zero dose. For instance, Bosaso city (which is the third most populous city in Somalia), we have conducted pre-registration (enumeration) to count the number of children under one year and under five years living in the city. The aim was to get a target denominator for Bossaso city before a new urban immunization strategy was implemented. The previous target denominator (before the registration) was 13,000 children but this time we obtained 28,000 under one-year children”*.(Humanitarian health worker, KII)

Despite current gaps in health and immunization data, three population groups were found to have a high proportion of zero-dose and under-immunized children across the country. These are nomadic populations, internally displaced people in camps, and communities living in Al-shabaab-controlled areas. 

*“These [zero dose] children reside in rural regions, villages, remote areas, and IDP camps, but in order to find out how many there are in total, a census or survey must be conducted in the areas”*.(MoH official, KII)

*“In general, there are a large number of children in Somalia who are not vaccinated for a variety of reasons. The first group of people are those who live in rural areas or remote villages without a health centre where mobile teams are unable to travel. The second group consists of children who live in IDPs; many of these children have not had any vaccines even though they live in towns where most vaccinations are provided at facilities nearby. Some urban residents do not vaccinate their children particularly those with low socio-economic status while others refuse it on religious beliefs and cultural grounds”*.(MoH official, KII)

#### 3.1.1. Nomadic Populations

The nomadic people do not stay in fixed locations throughout the year. There are currently no official estimates of the proportion of children who live in nomadic areas. 

*“We do not know how many of these children there are in the nomadic groups, let alone the percentage of immunized and un-immunized children. Regular routine immunization delivered through fixed sites does not offer convenient access to immunization for these communities and outreach programmes specifically designed for them are non-existent”*.(MoH official, KII)

Delivering immunization services to pastoralist areas requires resources and capacities which the government believes they could not afford.

*“Some of the challenges to expanding immunization services to rural areas [where nomadic populations reside] are related to resources and electricity, while others are related to security and geographic accessibility”*.(MoH official, KII)

The reported rate of zero-dose children in the nomadic communities is based on the population estimates from the 2014 Population Estimation Survey of Somalia (PESS) and SHDS 2020. When asked about the methods to estimate the rate of immunization coverage in nomadic communities, an MoH official explained,
*“We have not conducted specific surveys to estimate and identify these children in pastoralist communities. We classify the great majority of them as un-immunized not only because the data on these communities is limited but there are no immunization services for them that can report coverage data”*.(MoH official, KII)

The vaccinators in the FGD pointed out that immunization service availability was one of the reasons that the nomadic communities were missed. A vaccinator said,
*“I think if all the small villages across the country had primary health units which could provide routine immunization services, more children in the nomadic communities could get vaccinated. The outreach programmes do not offer routine immunization and only become popular response when there is an outbreak of measles, cholera or during polio national immunization days”*.(Vaccinator, FGD)

#### 3.1.2. Internally Displaced People (IDPs)

Unlike nomadic populations, IDPs reside in fixed formal or informal camps in and around major towns and cities where there is better provision of and access to basic health services, including immunization. While this community is identified as having a high proportion of unvaccinated children, the actual number is unknown. An NGO staff member said,
*“The IDP population changes constantly. New arrivals are received daily in some towns of the country. Most of the IDPs live in overcrowded informal temporary settlements which face threats of eviction. We estimate the number of households in each camp through the data captured in the provision of humanitarian assistance [food and cash assistance] but the actual number of people and composition is not known, particularly the number of children in each household. Some IDPs move between camps in the same town or to other parts of the country. We identify the IDP children as under-immunized because most of them originate from areas [rural villages, nomadic areas and locations controlled by Al-shabaab] where there are not regular immunization services”*.(NGO worker, KII)

An MoH official observed that the data on immunization coverage among IDP children living in camps are improving. He said,
*“Many IDP camps have fixed health centres or are located in the vicinity of health facilities where they share with host communities. Despite, lack of disaggregation of the rate of coverage between IDP and host communities, it is possible to analyse the data from health centres in IDP camps”*.(MoH official, KII)

The generalization of IDPs as a zero-dose community could be problematic as their vulnerability depends on where they originally resided, the length of their residence in the camps and the location of the camp, as one vaccinator in the FGD described,
*“People who have been living in an IDP camp for a long time may vaccinate their children, but zero-dose children are often from families who migrate to a new city every month, and those who originate from inaccessible places in Somalia”*.(Vaccinator, FGD)

Immunization service availability, particularly the lack of service provision at convenient times, is described as a major barrier for the IDP population, affects the demand and access for immunization, and contributes to a large number of zero-dose children in this community. 

*“IDPs live in major cities where they earn their livelihood, where there are good health facilities. Their challenges stem from their livelihood. They are casual labourers (manual labour). They leave home early in the morning and come back in the evening sometimes late but the working hours of the health facilities (health workers) are between 7 a.m. to 2 p.m. therefore there is missed opportunity here which is very high. They take some of the children with them and leave others at home. The children who are left at home cannot be immunized during the outreach because of lack of parental consent. We cannot say that they have access to health services even though the health facilities are very close. For these families, immunization is not a priority for them, survival and earning an income (livelihood) is”*.(Vaccinator, FGD)

#### 3.1.3. Populations Living in Al-Shabaab-Controlled Areas

Study participants noted that large segments of the population living in the Jubbaland and Galmudug regions are inaccessible to reach due to conflict, security reasons, and fear of obduction of healthcare workers. These areas are identified to host a high proportion of zero-dose and under-immunized children. 

*“Large areas of Galmudug are inaccessible due to insecurity, preventing the government and partners from reaching the people living in those areas and providing basic social services. The majority of these places are home to unvaccinated children, and they are in large numbers, with Al-Shabaab militants controlling roughly 40% or 50%”*.(MoH officials, KII)

When an MoH official was asked about immunization plans and priorities for his State, he replied,
*“Of course, reaching unvaccinated children is a priority (zero dose) for us but where they live is a challenge; in remote areas where the government has no control. There are genuine security concerns here as well. There have been reports of workers being kidnapped and held captive for months”*.(MoH official, KII)

Referring to the limited health services, including immunization in areas controlled by Al-shabaab, a humanitarian worker said,
*“There are numerous areas that cannot be visited due to security concerns. Children receive immunizations in locations from Bulo-Gadud to Kismayo town (small area), but beyond that, about 30 km from Bar-Sanguni (a nearby village to Kismayo), there are no vaccination facilities at all. The situation of the rural population is much worse, which is why there are so many outbreaks”*.(Humanitarian health worker, KII)

The prospect of expanding access to health and immunization services to populations in Galmudug and Jubbaland is often linked with the liberation of the areas from Al-shabaab militants. 

*“The immediate and most important aspect is to focus on security. Secondly, to secure the resources to reach these destinations. Opening a new health centre is not an easy task, so it is appropriate during the transition time to temporarily facilitate service delivery, such as using outreach programmes or mobile teams. Additionally, social media, community gatherings, and meetings will be used to raise awareness on immunizations”*.(MoH official, KII)

#### 3.1.4. Other Missed Subgroups

The participants mentioned urban poor and minority groups to be among the missed communities with high zero-dose and under-immunized children. The participants described the diversity of this subgroup and how poverty, social norms and behaviours and lack of awareness could underpin their vulnerability and constitute the main barriers to their access to immunization services. 

*“It is not a choice for many poor parents to vaccinate their children or not. They have bigger and more urgent struggles in their life; finding work, earning livelihood, housing, ill-health and unlike IDPs, most of them do not get assistance from the humanitarian agencies. These groups are very diverse. They could be minority communities living together in deprived areas in towns and cities where there are no good health facilities or could be individual poor families spread across different quarters in towns and cities”*.(MoH official, KII)

*“Unvaccinated children can be found in a variety of settings, including cities, remote locations, and areas of conflict, particularly in areas where the government has little authority. The zero dose children may live in cities with parents who embrace inaccurate beliefs about vaccines, such as that immunizations result in infertility”*.(Humanitarian health worker, KII)

A vaccinator voiced her dissatisfaction with the limited social mobilization and engagement with the urban poor. She said,
*“Some of the poor families in the neighbourhood of this health centre do not bring their children to the health facility for immunization. I talked to one family who said that they were not fully informed of the benefits of vaccination. Since their children were healthy, thanks to Allah, they did not think they needed to bring them to the health canter”*.(Vaccinatr, FGD)

A humanitarian worker attributed the low uptake of immunization among certain neglected communities in Puntland to a lack of health and immunization services. He said,
*“I have recently visited the villages along the Indian Ocean in Karkaar and Gardafui regions (Puntland). I witnessed widespread deprivation and lack of basic health services for some of these communities, particularly the minority community in Garduush village. I have no doubt that the great majority of the children in these communities were not immunized”*.(Humanitarian health worker, KII)

The lack of accurate and complete data on immunization coverage has been a recurring theme in the majority of the interviews and FGD. A humanitarian worker who was involved in immunization programming said,
*“I have not seen any study which was specifically done on immunization service delivery and uptake for marginalized groups but when we were discussing about urban immunization strategy, we looked into the existing barriers to immunization in major cities and towns such as Galkacyo, Burtinle, Garowe and Bosaso. We did not differentiate the barriers that had affected particular groups but daily stressors [livelihood circumstances, poverty, distance] and lack of education and awareness were noted as common impediments to access to immunization services”*.(Humanitarian health worker, KII)

### 3.2. Theme 2: Reaching Zero-Dose Children and Missed Communities

#### 3.2.1. Existing Immunization Policies and Strategies

The MoH officials explained that the Health Sector Strategic Plan (HSSP III), Somali National Immunization Policy (NPI), and the Expanded Programme of Immunization (EPI) policy provided strategic direction and guidance for the coordination, management, planning and delivery of immunization services in Somalia. However, a key government official described the challenges of developing and implementing immunization strategies and services for unvaccinated children. 

*“In reality, the focus of the national immunization policy is on vaccination in general. There is a brief mention of “missing doses” (p. 20) which does not fit with the scale and gravity of unvaccinated children in the country. I think there is no other way except to start with a separate strategy that focuses specifically on children who have not received any vaccine. Because there are more children who haven’t yet been vaccinated, this special strategy should chart how to identify and reach them, without interrupting the ongoing routine immunization activities”*.(MoH official, KII)

Despite the increasing awareness about zero-dose and under-immunized children, the government’s lack of clear strategies and plans to address them was echoed by other participants.

*“I do not know how we are going to do it, but a unique programme should be developed to reach areas where zero-dose children are detected, such as communities without a health centre, people in IDPs camps, and rural communities. Let’s have a look at how to vaccinate the newly displaced people. I believe these issues should be prioritized because vaccine-preventable diseases (VPDs) can spread from one child with a zero dose to another”*.(MoH official, KII)

According to Somali health authorities, there is some evidence that the policymakers understand the unique circumstances of the nomadic and IDP communities in the country and the challenges of reaching them, as well as their willingness to experiment with new immunization delivery strategies and models. Puntland State MoH claimed that they established a unit within the EPI programme targeting the nomadic and IDP communities but suggested that it was inactive due to a lack of funds. 

Recently, the MoHs, with the support of WHO and UNICEF, piloted the urban immunization strategy in the three cities of Mogadishu, Hargeysa and Bosaso. A participant who was involved in developing and piloting the strategy said,
*“In the strategy, there are components that ensure zero dose children are reached such as pre-immunization registration (head count of children) in order to set an accurate and actual target denominator. Thus, you can know the target children (number of children) to be vaccinated each month during the course of the year. This strategy was abandoned due to lack of funds”*.(Immunization programme officer, KII)

#### 3.2.2. Local Participation in Planning and Decision Making

The participants commented on the participation of local NGOs in planning and decision-making spaces and argued that local organizations should have a greater role in designing local strategies and interventions to deliver immunization services as they have good knowledge and experience about the context and enjoy a strong relationship with communities. 

*“The Ministry of Health, WHO and UNICEF are at the forefront of planning and coordination. We are seen as the implementers. We take part in the health sector coordination meetings. We mainly contribute to discussions about community needs and service delivery issues but our leverage is limited when it comes to policy development and planning unless we are the ones paying for the work to formulate and implement such policies and plans”*.(A senior official of a local NGO, KII)

At regional and district levels, there appeared to be parallel structures and mechanisms for planning and decision making for vaccination services, in which local organizations were more engaged. 

*“In these meetings, discussions are focused on the progress of the immunization activities per the workplans, analysing the data on vaccination coverage and the challenges that have been encountered. If the coverage is low, then the cause is investigated, and an action plan is agreed to improve the situation and increase the coverage. This information is rallied to the health canters where the vaccination programme is delivered”*.(Community member and a former local NGO staff, FGD)

#### 3.2.3. Immunization Service Delivery (Availability and Accessibility)

When discussing service availability, participants noted that immunization services were provided in fixed public health facilities and by occasional mobile teams which travel to communities, villages, and IDP camps where there are no health facilities. To supplement the routine vaccination, campaigns are organized in different parts of the country to reach more people.

An MoH official highlighted the limited delivery approaches in EPI and said,
*“Either you live a place close to health facility where routine immunization is available or you are covered by occasional outreach programme. There is no other way you can get your children vaccinated particularly if you live in a remote location or places where the government is not present”*.(MoH official, KII)

Participants reported how the vaccine supply and cold chain in Somalia are managed and coordinated at different levels by UNICEF, FMoH, State MoHs, and partner agencies. Somalia has a national cold chain, which is currently located in Nairobi and managed by UNICEF. There are three zonal cold chain stores in Somalia, located in Mogadishu (South-Central), Hargeisa (Somaliland) and Garowe (Puntland). An MoH official summarized the limited role of the government in vaccine supply and cold chain. 

*“In Somalia, UN agencies manage resources [vaccines] from the point of purchase to the health centre where they are used, however, the government may be engaged in the planning. We, in Puntland, manage the central cold chain for the State, but UN agencies (WHO, UNICEF) provide all financial assistance, logistics, and personnel salaries. The government’s involvement is limited to planning, forecasting, and occasionally budgeting of operating costs, which they [UN] also manage [funds]”*.(MoH official, KII)

The participants described the disruptions caused by COVID-19 lockdowns to illustrate the fragility of Somalia’s vaccine supply chain and service provision. One MoH official said,
*“Everything came to standstill. Vaccine transportation has been halted due to flight restrictions. Our main vaccine store is in Nairobi, and there have been some delays, resulting in supply shortages. Additionally, the community’s connectivity was restricted, and people were advised to stay at home. As a result, vaccination uptake was extremely low because many people did not visit health centres or other locations where vaccinations were administered. It was really difficult time for us to conduct an immunization, supervision or any kind of work at the time”*.(MoH official, KII)

Other participants noted the impact of the COVID-19 pandemic on interrupting vaccination services at health facilities:
*“COVID-19 has impacted on all health services. Many people have been unable to come to the health centre for months, leading to a decrease in the number of people being served. The people themselves have been affected, and mothers refused to come and bring their children”*.(MoH official, KII)

#### 3.2.4. Resources and Capacities 

An MoH official cited inadequate resources and capacity as the main challenge in identifying and reaching zero-dose and under-immunized children in Somalia. He said,
*“WHO and UNICEF support our vaccination programme. We are dedicated to promoting immunization across the country. Our partners have an interest to see that through. We have rough idea about the sort of communities where zero-dose children live but to be honest, we need adequate resources. Our capacity is limited but with resources, we can adapt strategies specifically targeting zero dose children and we can build our capacities to develop and implement effective plans and interventions”*.(MoH official, KII)

The participants commented on the lack of a credible chain of accountability in relation to resource allocation and utilization. There was a broad agreement among the participants about the detrimental effects of scarce resources, capacity constraints and the dependency on donor funds. However, some participants argued that it was not only the inadequacy of funding but also the inefficient use of resources. One MoH official said,
*“I hear that donors are providing lots of money to immunization services. Those of us at the FMS MoH level do not have information about the total amount of all that money. But we must have some kind of accountability and transparency. We can start with correcting the balance in decision making, implementation and oversight. Not only do we need an inclusive authority to coordinate and allocate the resources, we must have independent body to scrutinize the utilization of the resources”*.(MoH official, KII)

Another MoH official attributed the resource constraints to poor planning and management. He said,
*“The immunization service should be equitably distributed. The health care professionals who perform immunization services and the communities should be consulted, and the few resources available should be allocated fairly and in accordance with the priorities and demands of each location”*.(MoH official, KII)

The participants described how COVID-19 had diverted global and country efforts from routine immunization:
*“The biggest impact of COVID-19 happened in 2021 and 2022, when regular immunization was diverted and the focus was shifted to COVAX, resulting in a drop in coverage in the Banadir region. The lockdown and the fear of the infection forced many people not to go to the health centers. People were being advised to keep away from one another, resulting in low coverage, and employees who were expected to work in EPI were focused on the COVAX vaccine”*.(MoH official, KII)

#### 3.2.5. Integration with Other Health Programmes

An integrated facility-based and essential outreach service, such as the co-delivery of nutrition and immunization services for rural communities or veterinary care with immunization services, was suggested as a model to improve the efficiency of service provision and potential approach to reach unvaccinated and missed communities. An MoH official explained that the issue of integrating immunization with the other services constantly comes up in meetings:
*“We all agree that it [immunization] is intertwined with other health services such as vitamin A administration and nutrition. A health card containing information about the different services given to the child such as immunization could be made mandatory to receive health services”*.(MoH official, KII)

This response was echoed by a humanitarian worker who noted,
*“It is absolutely essential to combine vaccination with other health services. For example, if a mother attends ANC, she is taught how to get her child vaccinated and the importance of vaccination. Besides that, the mother is given information about the health risks if the child is not vaccinated. As a result, other health services often help and increase the number of children reached. It is critical, and I advocate that other health services be linked to vaccination. As far as we can reveal, it is a good thing. If vaccination is not highlighted in all health service delivery, public and private, then the importance of vaccination will not be realized. If the mother of the sick child is informed of the advantages of the vaccine while she is receiving medical help, she will accept it and spread the message to others”*.(MoH official, KII)

An MoH official in Puntland shared how they have explored integrating immunization services and veterinary care for the nomadic population, but the strategy was shelved because of a lack of funds. 

#### 3.2.6. Community and Individual-Level Vaccine Uptake

The participants suggested that some of the main reasons for the low vaccine uptake in Somalia could be due to vaccine hesitancy among different groups in the community. 


*“Communities participate in immunization, but it is not enough. The scholars participate, young women and educated people vaccinate their children but there is a lack of ownership, or a lack of direction and urgency for taking ownership of immunization. I think there is a need for government—community partnership. On the other hand, those who are misinformed, such as the Takfir religious group who believe that vaccines are not safe and question their permissibility in Islam, need a different approach, like active engagement and well-planned community led, religion-based strategy to fight misinformation”*
(Vaccinator, FGD)

Participants noted that while misconceptions about vaccine safety and risks are real issues that need to be addressed, it is simplistic and unhelpful to attribute the cause of the multifarious forms of vaccine hesitancy merely to misinformation and individual-level decisions. One participant in the FGD laid strong emphasis on trust and described the important role of well-known community members in awareness raising for immunization programmes. 

*“People have different views on vaccines. One-person next door to the MCH may reject them, while another may go a long distance to have their child vaccinated. Some people reject it for no obvious reasons, while others reject it because of incorrect information they have received from others or worries about harm on their children or they do not believe in its benefits. Perhaps those who refuse vaccination would listen to and share their private worries or reasons with people they trust such as midwives and other prominent people in the community”*.

### 3.3. Theme 3: Monitoring and Measuring the Identification of Zero-Dose Children and the Progress of Immunization Interventions

The participants did not provide details of specific strategies to measure or monitor zero-dose and under-immunized children, but they described the overall data gathering, management, and reporting system of immunization services in the country. Most of the participants agreed that the main challenge in monitoring and measuring immunization coverage was related to the lack of accurate demographic data to determine those who were unvaccinated. Official statistics such as civil registrations do not exist in Somalia. PESS 2014 and SHDS 2020 serve as the baseline for vaccine coverage and other health services in the country. 

An MoH official gave an example of how they calculate the immunization coverage rate. He said,
*“The MoH has a master list of all health facilities. Each health facility is designated to a catchment area and population. The population size of each area was calculated by adding the estimated data from the PESS 2014 with a 2.5% annual increment”*.(MoH official, KII)

He continued to describe the enormous challenges in establishing effective monitoring and measuring systems for immunization in Somalia, where there are such data gaps. He said,
*“All our sources of official data are poor. Civil registrations do not exist. The population estimate [PESS 2014] was conducted a long time ago, at a time when most of the South-Central Somalia was controlled by Alshabaab and the health and demographic survey [SHDS 2020] did not provide us concrete information about population sizes and compositions in different regions and communities. This is a cross-departmental issue and affects all basic social service delivery. However, we have now improved the reporting of health data including immunization coverage through our HMIS”*.(MoJ official, KII)

A vaccinator in the FGD noted the impact of a lack of accurate information in her area. 

*“We report the number of children we immunized and the MoH calculate the coverage rate. To be candid about the immunization data, when we do not have accurate number of children in our area or who they are, we cannot measure performance and cannot set targets for increasing the uptake. Our immunization coverage rate is actually more of reporting requirement than useful tool for planning”*.(Vaccinator, FGD)

Another MoH official described the difficulties of measuring immunization coverage in Somalia, particularly populations on the move. 

*“The target is children under nine (9) months old, and vaccination is planned based on centre needs. In addition to the lack of a specific target or precise microplan, the challenge arises from not having accurate statistics on vaccinated children due to continuous immigrant [IDPs, nomadic population] influx”*.(MoH official, KII)

MoH officials explained how they verified immunization data on the reporting tally sheets from health facilities. As an MoH official puts it,
*“If the Maternal and Child Health (MCH) clinic claims to have vaccinated 500 children, this information should be on the contact list, and their details [parent telephone numbers] should be in the registries. I’ve personally dialled random numbers to cross-check. There’s a system in place to verify the accuracy of the provided information”*.(MoH official, KII)

An immunization officer shared how they calculate the dropout rate of children who started the immunization by subtracting the number of children receiving Penta 1 from those receiving Penta 3. He said,
*“Vaccinated children from PENTA 1 and PENTA 3 are compared, assessing dropout rates and taking steps to understand access and utilization”*.(Immunization programme officer, KII)

Despite challenges with data completeness and accuracy, MoH officials reported that they improved the quality and use of available health data by reporting and analyzing the data via the digital platform DHIS-2. However, they acknowledge a lack of statistical skills to generate advanced predictions, associations and models. According to MoH officials, they hold technical working group meetings to discuss ways to enhance immunization coverage. 

*“Vaccination-related information is indeed utilized. Regular meetings, such as our technical working group meeting (TWG), occur alongside immunization meetings at district and regional levels. These sessions evaluate vaccine centre operations, administration, development, and community coverage. If coverage is low, the cause is identified, methods to enhance it are explored, and solutions are sought”*.(MoH official, KII)

### 3.4. Theme 4: Engagement and Advocacy Approach for Reaching Zero-Dose Children

#### 3.4.1. Political Leadership and Accountability

Immunization advocacy governance in Somalia follows the same pathway as the immunization programme leadership and coordination, which was discussed in theme 2 above. The Federal MoH (FMoH) is responsible for coordinating advocacy work with UNICEF and WHO at the national level, while the MoH of the Federal Member States (FMS MoH) coordinates advocacy and community engagement with implementing partners and State level representatives of UNICEF and WHO. MoH officials repeatedly pointed out the dominant position of WHO and UNICEF in Somalia’s National Health Cluster and the technical working group for immunization. An MoH official alluded to the hierarchy of coordination efforts among the multiple external and internal actors in the health and immunization governance and service delivery. He said,
*“We hold regular Health Cluster meetings. WHO, UNICEF, UNFPA and, to lesser extent, the International NGOs dominate this forum and set the agenda. Although the information about the meetings is publicly available, they are not more than progress updates and the cluster became a programming stage to prevent overlap of interventions among the UN and International organisations”*.(MoH official, KII)

Political leadership and accountability for advocacy were viewed through different lenses. While most of the participants from the MoHs argued that there was a high level of political leadership and strong support for immunization, some of the community members and vaccinators were skeptical.

*“We cannot judge commitment just by the occasional presence of top government officials in immunization events. We want them to make promoting immunization every day task which feature in all events. We want them to engage religious Sheikhs and raise awareness in the mosques. We want then to engage elders in the rural communities and propagate the messages in community gathering in the villages. And above all, we want them to hold the MoH and partner organizations accountable for this life-saving service”*.(Vaccinator, FGD)

However, some participants made more positive comments in support of the government’s advocacy functions. A community member in the FGD said,
*“All the immunization campaigns are launched by the ministries of health, and some instances by the FGS [Federal Government of Somalia] president and FMS [Federal Member States] presidents. This public show of support is very important for the immunization and underscore the leadership understanding and commitment for these preventative services”*.(Community member, FGD)

Another community member in the FGD was unconvinced of the launching of the immunization campaigns by politicians and expressed the need for a more practical show of support and action. 

*“I think it would send a powerful signal of support if the top political, clan and religious leaders were seen taking their children to health facilities for immunization, not just a minister cutting ribbon or putting vaccine drops into a child’s mouth for the start of immunization campaign”*.(Community member, FGD)

#### 3.4.2. Partnership and Collaboration for Advocacy

The participants described engagement and collaboration among different stakeholders via the Health Cluster and other State and regional coordination meetings. However, decision making was not evenly distributed but rather skewed towards the higher level where the FMoH, WHO, and UNICEF dominate power and resources. One humanitarian worker noted,
*“UNICEF and WHO wield huge influence because funds for the immunization programme are channelled through them and they have lots of experience and expertise at their disposal. With money, you can hire specialists and experts in the field”*.(Humanitarian health worker, KII)

The participants mentioned a network of trained social mobilizers under the Somalia Social Mobilization Network (SOMNET), which was deployed across the country but did not provide more information about the governance structure, financing and role of the network. An evaluation report on the network in 2017 detailed the role of social mobilizers in community engagement, community risk level communication, social mobilization, and health promotion at the national, state, regional and district levels [25]. The network is financially supported by GPEI via UNICEF but is used for all immunization social mobilization activities and other public health and nutrition promotions. 

#### 3.4.3. Community Engagement

Participants reported that healthcare workers and social mobilizers engage with the community and provide health education and awareness on immunization. One of the MoH officials noted,
*“Each village has a Community Health Committee (CHC) that educates people and reports on community needs. In addition, throughout health projects, we use community health workers (CHWs) who connect the community and health centres and participate in areas where vaccinations are administered to bring people together. At times, such as World Immunization Day, the community members are called to meetings to educate them on the benefits of vaccination. CHC comprise of respected individuals in the community (scholars, midwives, and elders) in order to persuade the people of the importance of vaccination”*.(MoH official, KII)

Participants in the FGD shared examples of various engagement approaches and interactions with the community, including establishing regular contact with community leaders and elders, local government officials and businesspeople. They stressed the importance of knowledge about power structures in the community and the selection of individuals with such knowledge and good standing in the community to be social mobilizers. 

*“The majority of advocates are young individuals, particularly females. It would be beneficial to involve other prominent and influential people in the community, such as elders so that they (social mobilizers) have better patronage to inform and persuade the community”*.(Vaccinator, FGD)

*“The MoH and implementing agencies need to understand that social mobilization and service delivery cannot be decoupled. The social mobilizers must be integrated with the health facilities and led by healthcare professionals who are respected in the local community. I think it would be helpful if local governors and community leaders sometimes deliver the messages”*.(Community member, FGD)

Despite the significant number of social mobilizers supported by UNICEF, one respondent had the view that they were not sufficient:
*“The people who work on raising awareness are very few in comparison to the community and size of the areas to be served. Awareness is the first and most important step to sensitize the community and promote acceptance. Hiring people who the community can trust and is willing to learn from them is very crucial if the awareness raising has to succeed”*.(Vaccinator, FGD)

## 4. Discussion

This study answers the questions of *who* and *where* zero-dose children and missed communities in Somali are, and *what* the strategies to effectively reach these population groups are. Nomadic populations, internally displaced persons in camps, and communities living in remote and Al-shabaab-controlled areas are three population groups with high proportions of zero-dose and under-immunized children in Somalia. Nomadic communities represent a significant percentage (approximately 26%) of the Somali population [10,12]. The exact number of zero-dose children in these communities is, however, difficult to estimate, as there is currently no data from censuses or demographic surveys specifically designed to account for this population group. Analysis of the SHDS 2020 shows that less than 1 percent of children living in nomadic areas were fully immunized against vaccine-preventable diseases [12]. Abdullahi et al. (2020) identified the nomadic nature of a large percentage of the Somali population in constant search of pastures and water, poor infrastructure in rural areas and distance to services as some of the main barriers to childhood immunization in the Galkayo district [26]. The frequent mobility of the nomadic population was associated with challenges to health service delivery, creating missed opportunities and under-vaccination [27,28,29,30,31]. However, the participants in this study warned against overemphasizing the herding lifestyle of the nomadic population as a problem since their pattern of seasonal movement and health-seeking behaviour is often predictable and known to local communities as well as health authorities. Rather, the complete absence of targeted and population-specific strategies and services was understood as a significant bottleneck in the health system. The study findings reveal that targeted vaccination strategies for nomadic populations have been drafted in Puntland but have yet to be implemented. Evidence from the limited available literature on immunization services for nomadic populations shows that innovative co-delivery approaches that integrate animal and human vaccination services could increase coverage and offer potential opportunities to reach unvaccinated children among these communities [32,33,34].

Similarly, the vaccination status of an estimated 3.8 million IDPs in the country living in and around large urban centres are unaccounted for, as there are currently no official data on the number of unvaccinated and/or under-vaccinated children among these missed communities in Somalia, not even in the recent SDHS 2020 which otherwise captures a broad range of health indicators [35]. This challenge of lack of accurate data impedes efforts to reach zero-dose children and measure immunization performance for these communities. A recent study has reported the complex and multi-layered barriers to childhood immunization ranging from limited knowledge and information about the types and schedules of immunizations, illiteracy, fear of side effects, vaccine stock-outs and distance from health facilities to cultural practices and the dominant position of men in decision making among IDP populations in Somalia [36]. The evidence from this study shows that health facility hours of operation were inconvenient for many IDP caregivers who earned a livelihood in the morning when the health centres were open. Even if an argument is made about the availability of immunization services at health centres according to the Somali national immunization and EPI policies, service availability is not determined by the mere availability of physical resources but by measuring the extent of the interaction between the resources and its users which depends on the types, quality and access of the service offered, convenience, communication and other appropriate support [37]. 

The study findings further show the specific vulnerability of people who live in remote and rural villages controlled by Al-Shabaab and who entirely rely on local NGOs for service delivery. The population in these areas is very diverse, including agricultural and fishing communities, residents of villages and small towns and nomadic populations. Insecurity and the government’s inability to reach these communities created structural barriers affecting access, coverage and equity and resulting in high proportions of zero-dose and under-immunized children. 

These three groups, while different in their contexts, vulnerabilities, and needs, represent jointly a significant proportion of the Somali population and cannot be neglected. In the literature and Somali context, these communities are often referred to as a “hard-to-reach” group [38]. Several factors, such as remote location, displacement conditions, poverty, insecurity and other vulnerabilities, are found to play significant roles in preventing vaccines from being available and accessible to these hard-to-reach communities and other marginalized communities [22,39,40]. 

This study illuminated the pervasive health and immunization data gaps in Somalia. The availability of reliable official health data in Somalia remains critical, without which equitable access to essential and life-saving health and immunization services will continue to be challenging. The sources of official statistics are limited, and civil registration of births and deaths is non-existent in Somalia. Some of the study participants questioned the completeness and reliability of the PESS 2014, SHDS 2020 and the administrative coverage data from health facilities, which serve as a baseline and method of estimating the rate of vaccination coverage in the country. These immunization data challenges could be addressed with strengthened health data collection and reporting, research and immunization coverage surveys that can generate the evidence base to design better-targeted strategies [6]. 

This study demonstrated the importance of considering the diversity of context, subgroup characteristics and social dynamics in the development of policies and delivery strategies which can boost immunization uptake among these communities. It has reinforced the argument that understanding the political, social and economic barriers of these subgroups will help avoid the mistakes of adopting a hegemonic view of one-size-fits-all policies and approaches for discrete contexts, locations, communities, and States [22]. 

Strong evidence exists that individual, community and health system level factors such as wealth, maternal education, proximity to health facilities, availability of services and vaccination governance have close associations with immunization coverage in resource-constrained settings [39,41,42,43,44]. Governance is particularly important as one of the main health systems’ building blocks. Ikilezi et al. (2022) have highlighted the association of governance with better utilization of development assistance and improvements in determinants such as maternal education and reduction in inequality that can have a positive effect on immunization coverage [45]. Cherian et al. (2020) have described several necessary conditions, such as a strengthened health system, country ownership and governance, evidence-based policies and strategies for any national vaccination programme to improve the quality and coverage of service delivery [46]. 

It is apparent that the current vaccination policies, delivery strategies and programmes in Somalia do not reflect the unique population demographics and health needs of the populations despite the evidence that context-specific strategies and targeted interventions could offer better approaches and higher chances for locating and engaging the identified subgroups, maximize coverage and promote equity [6,41]. However, without adequate resources and capacities, designing and implementing such strategies and delivery models will remain more challenging.

Targeting zero-dose children and missed communities with better delivery strategies and services extends beyond increasing immunization coverage rates and addresses some of the health disparities between communities and locations. However, to reach these communities requires doing things differently. Better governance and coordination at different levels, adequate resources, using alternative routes of service delivery to communities where the government cannot reach, generating demand with understandable vaccine information, and more robust evaluation methods are several options that should be considered. 

The Somali health system focuses mainly on mother and child health care services [7,47]. Upon close examination of the study participants, we observed that there were more female frontline health workers (vaccinators) in the health facilities and male-dominated MoH senior positions. Despite the lack of openly available data on the gender distribution of Somalia’s healthcare workforce, the all-male senior MoH officials and NGO staff in the study KII shed some light on gender disparities in senior leadership positions and indicate that Somali women’s voices and perspectives are missing in the higher policy and decision-making spaces in the Somalian health system.

In Somalia, prioritization and local know-how remain critical in reducing vulnerability and delivering equitable child health and immunization services for marginalized and neglected population groups. There is a huge untapped potential of human resources for health in the country with young and competent health cadres from both local communities and the diaspora. Unlocking creatively local capacity and know-how for health may increase resilience in the health system.

## 5. Conclusions

The three specific subgroups, nomadic populations, IDPs, and communities living in Al-shabaab-controlled areas, were identified in this study as having the highest proportions of zero-dose and under-immunized children. These groups, while differing in context and social dynamics, share some common vulnerabilities and barriers to childhood immunization, such as marginalization and challenges of access to health facilities and services. Despite the knowledge of these subgroups with the highest proportions of zero-dose children, there appears to be limited or no serious attempt to adopt context and sub-group-specific policies and delivery strategies to reach them with basic health and immunization services. The study concluded that improving access to childhood immunization for these specific subgroups will necessitate engaging with the communities, improving governance, building strong partnerships and collaboration to advocate for adequate resources, as well as developing targeted policies and delivery strategies which can address the main barriers to service availability and utilization. Establishing effective monitoring and evaluation systems and integrating the vaccination with other services such as nutrition, veterinary, humanitarian assistance and other outreach and support programmes could make these services closer to the missed communities and generate demand. Last but not least, improving the availability and quality of immunization data by strengthening the HMIS, conducting research and immunization coverage surveys will be critical for identifying where and how many zero-dose and under-immunized children are in Somalia and ensuring effective planning, delivery and evaluation of immunization interventions. 

### Limitations

The study interviewed senior policymakers in the MoHs who were involved in the day-to-day planning and programming of immunization services in the three study sites. The main limitations of the study arose from the study scope and timeframe. There was not sufficient time to investigate in-depth the socio-cultural, religious, gender based and economic barriers and other inequalities that afflict missed communities in the study sites. Because the focus group discussions were limited, it was not possible to assess the access to immunization services for other hard-to-reach communities such as minority, marginalized and religious groups that could have high rates of zero-dose children. We could not recruit participants from nomadic communities for the focus group discussion due to time and logistical constraints as these households are dispersed across large areas.

## Figures and Tables

**Figure 1 vaccines-12-00154-f001:**
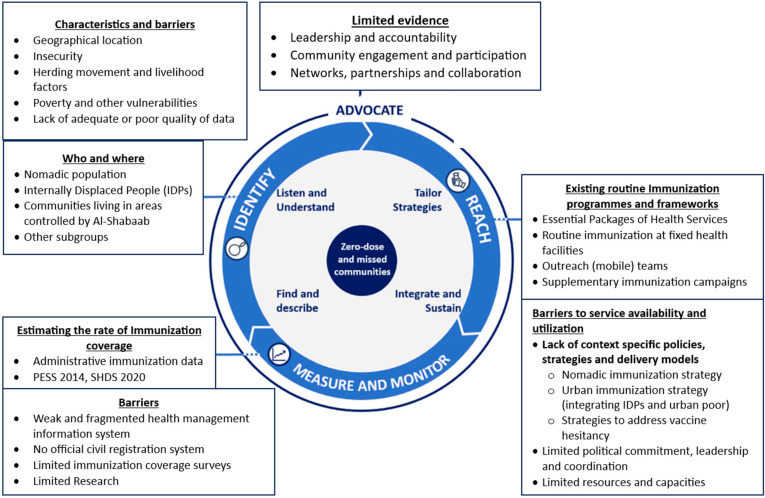
Study findings mapped against the IRMMA framework.

**Table 1 vaccines-12-00154-t001:** The five steps of Gavi’s IRMMA framework for zero-dose and under-immunized children.

Step	Description
Identify	This first step describes the strategies and methods to gain a “clear understanding of which, where, why, and how many zero-dose children and missed communities have not been reached”, including how they were identified (data sources) and the barriers that prevent these children from getting vaccinated [17].
Reach	The second step outlines the whole health system and primary healthcare approaches that countries can adopt, using a wide range of data and evidence, to design context-specific, tailored and sustainable delivery strategies and models which address the main barriers to service availability and service utilization for zero-dose and under-immunized children.
Monitor/Measure	The third and fourth steps describe the importance of effective monitoring and evaluation mechanisms to track and review progress in a timely manner and gain a better understanding of the extent to which the strategies and programmes are implemented in order to enable learning, course corrections and validation of the integrity of the implementation.
Advocate	The fifth step sets out approaches that governments can raise the profile of immunization, engage with affected communities and establish strong networks and partnerships in order to mobilize and prioritize resources towards zero-dose children and missed communities.

**Table 2 vaccines-12-00154-t002:** Key characteristics of participants in the key informant interviews.

	Variables		Number (*n*)	Percentage
Age Groups	19–29		5	29%
	30–40		10	59%
	40+		2	12%
Gender	Male		17	100%
	Female		0	0%
Agency	Federal MoH		3	18%
	State MoH	Puntland	3	18%
		Galmudug	2	12%
		Jubbaland	3	18%
	UNICEF		1	6%
	INGOs		3	18%
	Local NGO		2	12%

**Table 3 vaccines-12-00154-t003:** Key characteristics of participants in focus group discussions.

MoH Level	Vaccinators	Community	Total FGD
Puntland	1 (2 male, 9 female)	0	1 (11)
Galmudug	1 (3 male, 9 female)	1 (2 male, 6 female)	2 (20)
Jubaland	1 (1 male, 11 female)	1 (1 male, 8 female)	2 (21)
Total	3 (6 male, 29 female)	2 (3 male, 14 female)	5 (52)

## Data Availability

Quotations from the KII and FGD are presented in this publication. Redacted and anonymized KII and FGD transcripts are held in a secure database and can be requested from study authors.

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
