# Peer review of "Assessing Vaccination Delivery Strategies for Zero-Dose and Under-Immunized Children in the Fragile Context of Somalia"

_vaccines, 2024, doi:10.3390/vaccines12020154_

Round 1

Reviewer 1 Report

Comments and Suggestions for Authors

Ahmed Bile et al. describe in their article the socio-political circumstances why it is difficult to reduce the number of unvaccinated children in Somalia to a considerable degree. 

The authors describe precisely the regions they focus their attention on, the types of populations present in Somalia and the population type reasons why it is difficult to approach unvaccinated children in a systematic way. The data they collect to come to their conclusions is based on interviews with two different key groups: employees of the Ministry of Health and Non-Governmental organisations and discussion groups consisting of care givers in the field. 

The authors come up with a complex picture of the society and the different population groups (see figure 1) with a variety of reasons why under the specific circumstances at the side of interrogation the vaccination rates are difficult to improve. Sometimes the main reason is the nomadic character of the populations, sometimes the security threads caused by armed rebel groups, sometimes the lack of interest in the population itself which is more concerned with economic survival than caring for the vaccinations of their children.

The article concludes with some recommendations to improve the situation emphasising that there is not a single solution to a problem that has a variety of reasons. 

I think the article presents the situation in a well structured way. It is a rather lengthly article but I would not shorten the direct citations from the interviews. It might be possible to reduce the length of the introduction and discussion. Otherwise I think the article is publishable as it is. 

Author Response

Reviewer Comments

Response

Ahmed Bile et al. describe in their article the socio-political circumstances why it is difficult to reduce the number of unvaccinated children in Somalia to a considerable degree. 

Thank you for your valuable and positive feedback.

The authors describe precisely the regions they focus their attention on, the types of populations present in Somalia and the population type reasons why it is difficult to approach unvaccinated children in a systematic way. The data they collect to come to their conclusions is based on interviews with two different key groups: employees of the Ministry of Health and Non-Governmental organisations and discussion groups consisting of care givers in the field. 

Thank you for your valuable and positive feedback.

The authors come up with a complex picture of the society and the different population groups (see figure 1) with a variety of reasons why under the specific circumstances at the side of interrogation the vaccination rates are difficult to improve. Sometimes the main reason is the nomadic character of the populations, sometimes the security threads caused by armed rebel groups, sometimes the lack of interest in the population itself which is more concerned with economic survival than caring for the vaccinations of their children.

Thank you. We appreciate your valuable comment and the fact that we have managed to provide a clear picture of the complex context in which this research was conducted.

The article concludes with some recommendations to improve the situation emphasising that there is not a single solution to a problem that has a variety of reasons. 

Correct. Thank you for your valuable and positive feedback.

I think the article presents the situation in a well-structured way. It is a rather lengthy article but I would not shorten the direct citations from the interviews. It might be possible to reduce the length of the introduction and discussion. Otherwise, I think the article is publishable as it is. 

Thank you for your valuable and positive feedback.

Re: the feedback on the length of the paper, we have now edited the introduction and discussion sections, as per the reviewer’s suggestion. We hope the revisions made reflect the comprehensive findings, the interpretation of the quotations (context, subgroup characteristics, policies, delivery strategies) and the implications of these findings.

Reviewer 2 Report

Comments and Suggestions for Authors

This article identifies who and where zero dose children are in Somalia and what strategies are in use to reach them.

The background and study settings are well described. The methodology is clear and appropriate. The findings are well presented. The demographic characteristics of the KII and FGD participants are not described to the same extent i.e. there are no age data for the FGD. The discussion is relevant and engages with relevant literature.

I found the following to be a very minor detraction that may need editing for clarification:

- in line 19 the authors mention three geographically diverse regions of Somalia but in backets they list 4 sub populations instead of the 3 regions. 

- in line 38 the authors mention "the four basic childhood vaccinations" and diphtheria, tetanus, and pertussis (DTP). It is not clear what the 4 basic childhood immunizations are.

Comments on the Quality of English Language

The English quality is good but may need minor editing.

Author Response

Reviewer Comments

Response

This article identifies who and where zero dose children are in Somalia and what strategies are in use to reach them.

Thank you for your valuable and positive feedback.

The background and study settings are well described. The methodology is clear and appropriate. The findings are well presented. The demographic characteristics of the KII and FGD participants are not described to the same extent i.e. there are no age data for the FGD. The discussion is relevant and engages with relevant literature.

I found the following to be a very minor detraction that may need editing for clarification:

-   in line 19 the authors mention three geographically diverse regions of Somalia but in backets they list 4 sub populations instead of the 3 regions.

Thank you for your valuable comment. For clarification: the 4 subpopulations described in the paper, live in all 3 regions.  For example, internally displaced people live in all the 3 regions but are more concentrated in Jubbaland and Puntland regions. Thus, these subpopulations are found not only in these regions but across all the regions in Somalia which is why the study regions are considered illustrative (representative). We hope this clarification is acceptable.

-  in line 38 the authors mention "the four basic childhood vaccinations" and diphtheria, tetanus, and pertussis (DTP). It is not clear what the 4 basic childhood immunizations are.

Thank you for your valuable comment. This has now been addressed.

Comments on the Quality of English Language

-      The English quality is good but may need minor editing.

Thank you for your valuable comment. We hope the revisions made improve the quality of the paper.

Reviewer 3 Report

Comments and Suggestions for Authors

Introduction: the authors say "the four basic childhood vaccinations, diphtheria, tetranus, and pertussis (DTP) vaccines"... but they only mention three. Tuberculosis is probably missing.

Evidence on populations at risk of vaccine preventable diseases and barriers to vital vaccination services remain critical and urgent, especially in countries with complex health system challenges.

Author Response

Reviewer Comments

Response

Introduction: the authors say "the four basic childhood vaccinations, diphtheria, tetranus, and pertussis (DTP) vaccines"... but they only mention three. Tuberculosis is probably missing.

Thank you for your valuable comment. This has now been addressed.

Evidence on populations at risk of vaccine preventable diseases and barriers to vital vaccination services remain critical and urgent, especially in countries with complex health system

Thank you for your valuable and positive feedback.

Reviewer 4 Report

Comments and Suggestions for Authors

We thank the authors for developing such interesting and relevant research on a priority topic of global relevance in the context of eradication and elimination of diseases preventable by vaccination.

The research shows the descriptive results necessary as an initial step to delve deeper into the topic and to generate new studies that answer new questions.

The study design was appropriate because of the characteristics of the population analyzed and the topic to be investigated.

As a case study, where the background is little or almost non-existent on the subject, the observations are only about the ease of reading and that the results are summarized and concise.

The recommendations are as follows:

1) Be more concise and avoid repeating words. An example is that the word "Zero-Dose" is repeated 68 times from the title to the materials and methods part. This prevents easy reading

2) Reading is difficult because the paragraphs are long and complex.

3)In the introduction, it is necessary to have information on the impact of having countries with ZERO vaccination in the context of eradication and elimination of preventable diseases worldwide. This highlights the impact of the current research and generates more knowledge on the topic.

The objective of this study appears to be more specific than its general purpose.

The major problem of this study is the large amount of text, so it is necessary to optimize writing to improve the fluidity and cohesion of the text.

There are no further recommendations to the research; we consider that this could be a needed research effort to address and visualize these problems in these vulnerable populations and future problems.

Author Response

Reviewer Comments

Response

We thank the authors for developing such interesting and relevant research on a priority topic of global relevance in the context of eradication and elimination of diseases preventable by vaccination

Thank you for your valuable and positive feedback.

The research shows the descriptive results necessary as an initial step to delve deeper into the topic and to generate new studies that answer new questions.

Thank you for your valuable and positive feedback.

The study design was appropriate because of the characteristics of the population analyzed and the topic to be investigated.

Thank you for your valuable and positive feedback.

As a case study, where the background is little or almost non-existent on the subject, the observations are only about the ease of reading and that the results are summarized and concise.

Thank you for your valuable and positive feedback.

The recommendations are as follows:

1)   Be more concise and avoid repeating words. An example is that the word "Zero-Dose" is repeated 68 times from the title to the materials and methods part. This prevents easy reading

Thank you for your valuable feedback. We acknowledge and agree with your observation. The choice of the IRMMA Framework in the design and interpretation of results makes the paper rather “lengthy” with some risks of “repetitive” of the concept of zero-dose. We hope the revisions made to the paper improves the “readability” of the paper.

2)    Reading is difficult because the paragraphs are long and complex.

Thank you for your valuable feedback. We hope the revisions made improve the readability of the paper.

3)   In the introduction, it is necessary to have information on the impact of having countries with ZERO vaccination in the context of eradication and elimination of preventable diseases worldwide. This highlights the impact of the current research and generates more knowledge on the topic.

Thank you for your important and valuable feedback. We assess that the impact of zero dose children in terms of reducing vaccine preventable diseases (herd immunity, etc) has been extensively studied elsewhere. While there is currently a gap in identifying, and reaching zero dose children, particularly fragile and humanitarian settings. This is why we choose to focus on studying this matter in the context of complex and fragile health system of Somalia. We hope this clarification is acceptable.

4)    The objective of this study appears to be more specific than its general purpose.

Thank you for your important and valuable feedback. This study is a part of a large research with the broader objective. This article focused on answering the question of 1) who and where zero dose children and under-vaccinated populations live in Somalia, and 2) what are the vaccine delivery strategies and mechanisms currently in place. We hope this clarification is acceptable.

5)   The major problem of this study is the large amount of text, so it is necessary to optimize writing to improve the fluidity and cohesion of the text.

Thank you for your valuable feedback. The choice of the IRMMA Framework in the design and interpretation of the study makes the paper rather “lengthy”. We hope the revisions made to the paper improves the “readability” of the paper.

There are no further recommendations to the research; we consider that this could be a needed research effort to address and visualize these problems in these vulnerable populations and future problems.

We appreciate your important and valuable comments. Thank you so much.